# Effect of Dietary Supplementation of Hydrolyzed Yeast on Growth Performance, Digestibility, Rumen Fermentation, and Hematology in Growing Beef Cattle

**DOI:** 10.3390/ani12182473

**Published:** 2022-09-19

**Authors:** Nirawan Gunun, Ittipol Sanjun, Chatchai Kaewpila, Suban Foiklang, Anusorn Cherdthong, Metha Wanapat, Sineenart Polyorach, Waroon Khota, Thachawech Kimprasit, Piyawit Kesorn, Nipa Milintawisamai, Pongsatorn Gunun

**Affiliations:** 1Department of Animal Science, Faculty of Technology, Udon Thani Rajabhat University, Udon Thani 41000, Thailand; 2Department of Animal Science, Faculty of Natural Resources, Rajamangala University of Technology Isan, Sakon Nakhon Campus, Phangkhon, Sakon Nakhon 47160, Thailand; 3Faculty of Animal Science and Technology, Maejo University, Chiang Mai 50290, Thailand; 4Tropical Feed Resources Research and Development Center (TROFREC), Department of Animal Science, Faculty of Agriculture, Khon Kaen University, Khon Kaen 40002, Thailand; 5Department of Animal Production Technology and Fisheries, Faculty of Agricultural Technology, King Mongkut’s Institute of Technology Ladkrabang, Bangkok 10520, Thailand; 6Department of Biochemistry, Faculty of Science, Khon Kaen University, Khon Kaen 40002, Thailand

**Keywords:** hydrolyzed yeast, average daily gain, propionate, bacterial population, hematological parameters

## Abstract

**Simple Summary:**

Hydrolyzed yeast consists of β-glucans, mannan-oligosaccharides, nucleotides, peptides, amino acids, and other compounds. It is a potential source of prebiotics for alternative antibiotics in ruminants. The aim of this study was to determine the different levels of hydrolyzed yeast supplementation on feed utilization, rumen fermentation, hematology, and growth performance in growing beef cattle. The current findings indicate that supplementation of hydrolyzed yeast enhances the nutritional digestibility, rumen fermentation characteristics, and hematology. However, this did not affect the growth performance of growing beef cattle.

**Abstract:**

This experiment was conducted to assess the effect of hydrolyzed yeast (HY) on growth performance, nutrient digestibility, rumen fermentation, and hematology in growing crossbred *Bos indicus* cattle. Twenty crossbred beef cattle with an initial body weight (BW) of 142 ± 12 kg were randomly assigned to one of four treatments for 90 d in a randomized complete block design (RCBD) having five blocks based on a homogenous subpopulation of sex and BW. Cattle were fed with a total mixed ration (TMR) and supplemented with HY at 0, 1, 2, and 3 g/kg dry matter (DM), respectively. Supplementation with the HY did not change average daily gain (ADG), dry matter intake (DMI), and gain to feed ratio (G:F) (*p* ≥ 0.06). The addition of HY did not adversely affect nutrient intake (*p* ≥ 0.48), while the digestibility of crude protein (CP) increased quadratically (*p*
*=* 0.03) in the cattle receiving HY. The addition of HY did not affect rumen pH, but NH_3_-N concentration increased linearly (*p* = 0.02) in the cattle. The total volatile fatty acid (total VFA) increased quadratically (*p*
*=* 0.03) when cattle were fed with HY supplementation. The proportion of acetate decreased cubically (*p*
*=* 0.03) while propionate increased cubically (*p*
*=* 0.01), resulting in a decrease in the acetate to propionate ratio (*p*
*=* 0.01) when cattle were fed with HY supplementation. In addition, acetate was the lowest, but total VFA and propionate were the highest in cattle fed the HY at 2 g/kg DM. Butyrate increased cubically (*p* = 0.02) with the addition of HY. The protozoal and fungal populations were similar among treatments (*p* ≥ 0.11), but the bacterial population increased linearly (*p* < 0.01) with the addition of HY. Supplementation of HY did not influence blood urea nitrogen (BUN), red blood cells (RBC), hemoglobin, hematocrit, white blood cells (WBC), lymphocytes, or eosinophils (*p*
*≥* 0.10). However, monocytes and neutrophils increased linearly (*p* = 0.04 and *p* = 0.01, respectively) by HY supplementation. In conclusion, supplementation of HY at 2 g/kg DM promotes CP digestibility, rumen fermentation efficiency, and hematology but does not affect the growth performance of growing beef cattle.

## 1. Introduction

Farmers, feed manufacturers, and animal nutritionists are becoming highly interested in feed additives to improve the feed utilization, rumen microbial fermentation, health, and performance of their animals in tropical areas [1,2,3]. Antibiotics have been used for the prevention and treatment of disease in animals, plants, and humans [4]. However, a large portion of the antibiotics generated each year around the world are utilized for nontherapeutic purposes [5]. In addition, antibiotics have been utilized as growth promoters and feed enhancers, and not for the treatment of disease [6,7,8]. However, given the prevalence of antibiotic resistance and concern about its future global impact, research into ways to support antibiotic restriction is critical [9]. Because of this, natural products have become more important as alternatives to antibiotics that can be used to change rumen fermentation, animal health, and growth performance.

Hydrolyzed yeast (HY) is a relatively new feed material that can be obtained through different methods, the most common of which are autolysis and hydrolysis [10]. Autolysis of yeast (*Saccharomyces cerevisiae*) is a degradation process by activation of endogenous enzymes to solubilize cell components within the cell [11]. HY is the whole yeast residue remaining after the lysis process and contains β-glucans, mannan-oligosaccharides (MOS), nucleotides, peptides, and amino acids [10,12,13]. These polysaccharides prevent pathogenic bacteria in the digestive tract and interact directly with immune cells [9]. HY addition improves anti-bacterial immunity mediated by macrophages and neutrophils [14].

HY provides various growth factors, vitamins, and other nutrients to promote the development of rumen microorganisms, especially bacteria [15]. It has been found that yeast stimulates lactate-utilizing bacteria to use lactic acid [16] and enhances the cellulolytic bacterial population, thereby increasing the digestion of fiber [17,18]. Additionally, it also stimulates fermentation and improves rumen VFA production, especially propionate [19], which may enhance growth performance in ruminants. We hypothesized that the supplementation of HY (g/kg DM) enhances feed utilization, rumen fermentation, health status, and growth performance of beef cattle. The aims of this study were to investigate the effects of supplementation of HY on growth performance, digestibility, microbial population, rumen fermentation, and hematology in growing beef cattle.

## 2. Materials and Methods

### 2.1. Ethical Procedure

Animal care and experimental techniques were both approved by the Animals Ethical Committee of the Rajamangala University of Technology Isan (approval number 21/2564).

### 2.2. Animals, Treatments, and Experimental Design

The study was carried out at the beef cattle farm of the Faculty of Natural Resources, Rajamangala University of Technology Isan, Sakon Nakhon Campus, Phangkhon, Sakon Nakhon, Thailand. A 90 day feeding trial was conducted using a randomized complete block design (RCBD) to compare dietary hydrolyzed yeast (Hilyses^TM^, ICC, Sao Paulo, Brazil) supplementation at 0, 1, 2, and 3 g/kg DM. Twenty crossbred (Brahman × Thai native) yearling beef cattle, consisting of sixteen males and four females with 142 ± 12 kg of BW (mean ± SD), were grouped into five blocks according to sex and their homogenous BW, respectively. The animals within each block were assigned randomly to one of four dietary treatments. The total mixed ration (TMR) was prepared with 40% rice straw and 60% concentrate mixtures. Rice straw was chopped (4 cm length) by machine (JrFarm, Nakhon Ratchasima, Thailand). Chopped rice straw was mixed with concentrate ingredients to prepare TMR. The cattle were provided TMR (Table 1) *ad libitum*, targeting a refusal of 10% on an as-fed basis and fed twice a day at 08:00 h and 17:00 h. The cattle were kept in separate pens all the time with access to water and mineral blocks.

### 2.3. Data Collection and Sampling Procedures

Cattle were weighed at the initial BW, 30 d, 60 d, and final BW at 90 d, and ADG was calculated. Every morning, feed offered and refusals were recorded and collected for chemical analysis. At the end of the experiment, a digestibility test was conducted in which cattle were housed in individual pens for five consecutive days (86 to 90 d of the trial). Rectal sampling was used to collect fresh fecal samples (about 500 g). The composite samples of each cattle’s daily fresh feces were combined and then refrigerated at 4 °C. Feeds, refusals, and feces samples were dried at 60 °C, and ground (1-milimeter screen using Cyclotech Mill; Tecator, Hoganas, Sweden). The amounts of ash, ether extract (EE), CP [20], neutral detergent fiber (NDF), acid detergent fiber (ADF) [20,21], and acid-insoluble ash (AIA) in the samples were investigated. The AIA was used to calculate nutrient digestibility [22].

At 4 h after feeding on the final day of the experiment, 200 mL of rumen fluid was collected using a stomach tube attached to a vacuum pump. The first 100 mL of the ruminal samples were discarded to prevent saliva contamination. The samples were then put through four layers of cheesecloth and immediately measured with a portable pH meter. In addition, 1 mL of rumen fluid was diluted with 9 mL of formalin–saline solution. The numbers of bacteria, protozoa, and fungi were counted with a haemacytometer (Boeco, Hamburg, Germany) by the Galyean method [23]. The remaining samples of ruminal fluid were centrifuged at 16,000× *g* for 15 min at 4 °C and the supernatant was kept at −20 °C. The ruminal samples were thawed and then used to measure NH_3_-N (Kjeltech Auto 1030 Analyzer, Tecator, Hoganiis, Sweden) [24] and VFA using a gas chromatograph (GC 8890; Agilent Technologies Ltd., Santa Clara County, CA, USA) [25]. 

Blood samples from cattle were taken at the same time as ruminal fluid samples and were collected from the jugular vein. Each cattle had fresh blood withdrawn from their jugular vein in the amount of 10 mL. Each blood sample was kept in tubes containing EDTA for measuring BUN [26] and hematological parameters. The concentration of RBC, hemoglobin, hematocrit, WBC, neutrophils, lymphocytes, monocytes, and eosinophils were assessed using a hematological analyzer (BCC-3000B; DIRUI, Gungoren/Istanbul, Turkey).

### 2.4. Statistical Analysis 

All data were tested for normal distribution using the UNIVARIATE procedure in SAS software and subjected to analysis of variance using the GLM procedure [27]. The data were analyzed using the model *Yi* = *µ* + *αi* + *βj* + *εij*, where *Yi* is the dependent variable, *µ* is the overall mean, *αi* is the treatment effect (*i* = 1 to 4; HY supplementation at 0, 1, 2, and 3 g/kg DM), *βj* is the block effect (*j* = 1 to 5), and *εij* is the residual error. Orthogonal polynomial contrasts (linear, quadratic, and cubic) were used to estimate the effect of HY supplementation. Significant effects were identified at *p* < 0.05. When the contrasts were statistically significant, the effect was evaluated at the higher level, i.e., cubic, quadratic, and linear, respectively.

## 3. Results

### 3.1. Performance

The supplementation of HY did not affect weight and ADG from 0 to 90 d (*p* ≥ 0.06) (Table 2). The DMI and gain to feed (G:F) were similar among treatments (*p* ≥ 0.13).

### 3.2. Nutrient Intake and Digestibility

The nutrient intake was similar among treatments (*p* ≥ 0.48) (Table 3). The digestibility of CP increased quadratically (*p* = 0.03) with increasing HY supplementation, but the digestibility of DM, organic matter (OM), EE, NDF, and ADF were similar among groups (*p* ≥ 0.06) (Table 3).

### 3.3. Rumen Fermentation and Microbial Population

The rumen pH was not significantly different among treatments (*p* ≥ 0.22), while rumen NH_3_-N linearly increased (*p* = 0.02) when increasing the level of HY (Table 4). The concentration of total VFA was increased quadratically (*p* = 0.03) when cattle were fed with HY supplementation. The proportions of propionate (C3) were increased cubically (*p* = 0.01), while acetate (C2) and C2:C3 were decreased cubically (*p* = 0.03 and *p* = 0.01, respectively) by HY supplementation. Furthermore, total VFA, propionate was highest, while acetate was lowest in cattle fed with the addition of HY at 2 g/kg DM. The butyrate proportion was cubically increased (*p* = 0.02) by HY addition. The supplementation of HY did not influence the protozoal and fungal populations (*p* ≥ 0.11) (Table 5). However, the bacterial population linearly increased (*p* < 0.01) with the supplementation of HY.

### 3.4. Blood Urea Nitrogen and Hematological Parameters

There were no effects on BUN, RBC, WBC, hemoglobin, hematocrit, lymphocytes, and eosinophils (*p* ≥ 0.10) (Table 6), while neutrophils and monocytes were increased linearly (*p* = 0.01 and *p* = 0.04, respectively) by HY supplementation. 

## 4. Discussion

### 4.1. Performance 

Previous investigations into the effects of HY supplementation have been variable. Crossbred feedlot steers supplemented with 1 to 3 g/hd/d of enzymatically HY during high temperatures had improved ADG and DMI during 139 to 229 d of a trial, and it did not affect the first 139 days of the experiment [28]. Salinas-Chavira [29] also found a correlation between increased ADG and DMI from 224 to 336 d in feedlot steers fed with enzymatically HY plus yeast culture at 195 to 585 mg/kg DM. In contrast, the supplementation of HY at 1–3 g/kg DM for 90 d of a trial did not alter the ADG and DMI of growing beef cattle in the present study. Similarly, Pukrop et al. [30] reported that feeding HY at 13 g/hd/d for 56 d did not change ADG and DMI in feedlot cattle. The addition of HY at 4–7 g/hd/d in feedlot steers did not influence ADG and DMI for the final 105 d [31]. The inclusion of HY improved the digestibility of CP and VFA production, especially propionate, but did not affect the growth performance of cattle in our study. There are two plausible explanations for these results. Firstly, a short trial period for evaluating the effect of HY in the experiment. Secondly, the animals were not stressed enough during the added HY, as it did not change the feed intake and growth performance.

### 4.2. Nutrient Intake and Digestibility

The HY supplementation significantly increased the digestibility of CP in beef cattle. The HY product contains highly digestible protein, amino acids, and nucleotides. Hence, the addition of HY to the cattle enhanced the digestibility of CP. Additionally, it has been found that HY cell wall components provide a substrate for cellulolytic bacteria, stimulating their growth in the rumen [32]. Neubauer et al. [17] found that the HY increased the population of cellulolytic bacteria, particularly *Ruminococcus* spp. Lei et al. [33] reported that the digestibility of fiber was improved by yeast cell wall supplementation in beef cattle. In contrast, supplementation of HY did not affect the fiber digestion of beef cattle in the present study. Similarly, Salinas-Chavira [28] found that enzymatically HY supplementation did not affect fiber digestibility in feedlot cattle. Variable responses may be related to supplementation levels, yeast products, dietary ingredients, or animals [34,35].

### 4.3. Rumen Fermentation and Microbial Population

Maintaining a consistent rumen pH is important for healthy rumen ecology, fermentation, and microbial growth since ruminal pH is the main indicator of the rumen environment [36]. Yeast products have the potential to stimulate the growth of rumen bacteria, particularly lactate-utilizing bacteria, or slowing the degradation of starch in the rumen [37,38]. Neubauer et al. [17] suggested that the bacterial community and subsequent fermentation were most significantly modulated by the HY additive during times when the pH was low. The rumen pH range for all treatments was 6.8 to 6.9, and the optimum for rumen ecology was a pH of 6.5–7.0 [39]. The addition of HY had no effect on the rumen pH, which was consistent with the results reported by Salinas-Chavira et al. [28], who found that rumen pH was not affected by the enzymatically HY supplementation in feedlot cattle. Kröger et al. [40] reported that the addition of HY did not change rumen pH in nonlactating cows. One possible explanation for why the addition of HY did not affect ruminal pH is that growing beef cattle fed a TMR with a R:C ratio of 40:60 had little effect on rumen pH.

In the rumen, NH_3_-N is the primary nitrogen source for microbial protein synthesis [41,42]. The higher NH_3_-N concentration in beef cattle given HY was consistent with the findings of Oztürk et al. [43]; the addition of HY improved the rumen simulation technique’s NH_3_-N concentration (Rusitec). Oeztuerk et al. [44] found that HY stimulated the proteolytic activity of rumen bacteria, which in turn led to an increase in the concentration of NH_3_-N in Rusitec. In the current study, the greater ruminal NH_3_-N concentration can be attributed to the microbial breakdown of yeasts due to their high protein content. The concentration of rumen NH_3_-N at 4 h after feeding ranges from 15.5 to 18.3 mg/dL, which is close to the optimum level (13 to 24 mg/dL) for microbial growth in the rumen [42,45,46,47].

The VFA serves as their primary source of metabolizable energy, resulting in an increase in ruminant production [19]. Other studies have found no effect on VFA production in the rumen of ruminants fed HY [28,43,44]. In contrast, the addition of HY improved propionate and reduced acetate and C2:C3 in an in vitro study [19,48]. In the present study, supplementation of HY shifted the rumen VFA profile to cubic increased propionate while cubic decreased in acetate, leading to an increase in the glucogenic potential [35]. This could be due to the nucleotides, peptides, amino acids, vitamins, and minerals of HY promoting the growth of the microbial population in the rumen and also improving the energy utilization of feed for propionate production. HY was added at 2 g/kg DM, with the highest propionate and lowest acetate proportions. This suggests that adding HY at 2 g/kg DM is suitable for the stimulation of the growth of microbes and also enhances propionate production in the rumen. Furthermore, supplementing with HY increased butyrate production. This could be because the addition of HY increased the number of *Butyrivibrio* spp., which are known to break down hemicellulose in the rumen and also produce more butyrate [49].

Dietary HY supplementation increased the bacterial population in the rumen of beef cattle fed TMR. The micronutrient in HY may stimulate the growth factor for bacteria in the rumen. Furthermore, HY can promote rumen maturation and a stable ruminal pH, allowing rumen bacteria to grow [50,51]. Most rumen bacteria can use ammonia as their primary nitrogen source [52]. High NH_3_-N concentration while adding HY can boost the bacterial population due to rumen biosynthesis.

### 4.4. Blood Urea Nitrogen and Hematological Parameters

The BUN concentration is commonly used to evaluate protein availability and metabolic problems associated with animal disorders [53]. The inclusion of HY did not affect BUN concentration and it ranged from 17.4 to 18.4 mg/dL, which was within the usual range of 10 to 20 mg/dL [53,54]. Hematological analysis has been used to assess the health and nutrition of animals. Supplementation of HY did not influence hemoglobin, hematocrit, RBC, WBC, lymphocytes, or eosinophils. However, neutrophils and monocytes were increased with HY supplementation. Similarly, Adili et al. [55] reported that neutrophils were increased by the addition of HY to dairy cows. Neutrophils can protect livestock against the most common infectious diseases [56]. Kim et al. [57] observed that Holstein calves fed HY showed enhanced neutrophils. Similarly, Wang et al. [14] indicated that live yeast increases the expression of genes that improve the function of neutrophils, especially those that code for the IL-4 receptor and IL-1B in dairy cattle. Pedro et al. [58] found that Dectin-1 activation increases the expression of pro-inflammatory cytokines in monocytes in response to β-glucan in yeast products. In addition, modulation of monocyte activation has also been related to bovine neutrophil degranulation [59]. These results indicate that the addition of HY to the cattle has the possibility of reducing inflammatory factors via enhanced neutrophils and monocytes in growing beef cattle. Previous studies have shown that the amounts of hemogram indices in cattle’s blood are within the normal range [54,60,61,62].

## 5. Conclusions

Dietary supplementation of HY enhanced the digestibility of CP and hematological indices, especially neutrophils and monocytes in growing beef cattle. In addition, supplementation of HY enhanced total VFA and propionate production. However, supplementation of HY did not affect the growth performance. The HY was fed to growing beef cattle at a dose of 2 g/kg DM as a rumen modifier and to improve health status. Therefore, further studies are required to determine the effect of HY on carcass traits and meat quality in beef cattle.

## Figures and Tables

**Table 1 animals-12-02473-t001:** Ingredients and chemical composition of the diet used in the experiment.

Item	TMR
Ingredient, kg dry matter (DM)	
Rice straw	40.0
Cassava chip	30.0
Rice bran	14.0
Soybean meal	10.0
Urea	2.0
Molasses	2.0
Minerals and vitamins	1.0
Sulfur	0.5
Salt	0.5
Chemical composition	
Dry matter, %	66.4
Organic matter, %DM	91.8
Crude protein, %DM	13.8
Ether extract, %DM	1.0
Neutral detergent fiber, %DM	37.9
Acid detergent fiber, %DM	19.7
Ash, %DM	8.2

TMR: total mixed ration.

**Table 2 animals-12-02473-t002:** Effect of HY supplementation on growth performance in growing beef cattle.

Item	HY (g/kg DM)					Contrast		
	0	1	2	3	SEM	L	Q	C
Body weight, kg								
Initial	128.8 (±19.97)	154.8 (±34.39)	136.4 (±18.79)	140.6 (±33.48)	4.13	0.61	0.17	0.07
30	150.2 (±23.71)	177.6 (±42.67)	155.2 (±21.32)	163.8 (±44.32)	4.49	0.65	0.32	0.06
60	165.0 (±24.15)	190.8 (±39.92)	171.4 (±23.95)	180.2 (±43.88)	4.46	0.52	0.36	0.09
Final	178.6 (±19.64)	204.6 (±45.75)	188.0 (±28.57)	192.8 (±42.96)	4.54	0.54	0.27	0.15
ADG, kg/d								
0 to 30 d	0.7 (±0.14)	0.8 (±0.34)	0.6 (±0.22)	0.8 (±0.37)	0.51	0.97	0.53	0.20
31 to 60 d	0.5 (±0.09)	0.4 (±0.17)	0.5 (±0.24)	0.5 (±0.15)	0.64	0.45	0.69	0.47
61 to 90 d	0.4 (±0.19)	0.5 (±0.34)	0.6 (±0.35)	0.5 (±0.15)	0.53	0.72	0.42	0.51
0 to 90 d	0.5 (±0.06)	0.6 (±0.18)	0.6 (±0.25)	0.6 (±0.14)	0.38	0.94	0.93	0.88
DMI, kg/d								
0 to 30 d	4.8 (±0.82)	4.9 (±1.28)	4.8 (±0.91)	4.7 (±1.06)	0.81	0.73	0.56	0.82
31 to 60 d	4.5 (±0.72)	5.0 (±1.02)	4.4 (±0.84)	4.9 (±0.73)	0.71	0.70	0.99	0.41
61 to 90 d	4.8 (±1.13)	4.9 (±1.29)	5.0 (±1.07)	5.0 (±1.38)	0.82	0.78	0.70	0.83
0 to 90 d	4.7 (±1.14)	4.9 (±0.74)	4.7 (±0.41)	4.9 (±0.81)	0.78	0.83	0.66	0.63
G:F								
0 to 30 d	0.2 (±0.03)	0.2 (±0.06)	0.1 (±0.03)	0.2 (±0.05)	0.19	0.95	0.37	0.28
31 to 60 d	0.1 (±0.02)	0.1 (±0.06)	0.1 (±0.04)	0.1 (±0.04)	0.17	0.66	0.95	0.48
61 to 90 d	0.1 (±0.05)	0.1 (±0.07)	0.1 (±0.05)	0.1 (±0.07)	0.23	0.57	0.61	0.62
0 to 90 d	0.1 (±0.03)	0.1 (±0.03)	0.1 (±0.03)	0.1 (±0.03)	0.13	0.61	0.68	0.51

**Table 3 animals-12-02473-t003:** Effect of HY supplementation on nutrient intake and digestibility in growing beef cattle.

Item	HY (g/kg DM)					Contrast		
	0	1	2	3	SEM	L	Q	C
Nutrient intake, kg/d								
Organic matter	4.3 (±0.86)	4.6 (±1.13)	4.4 (±0.84)	4.4 (±1.06)	0.13	0.82	0.66	0.63
Crude protein	0.7 (±0.13)	0.7 (±0.17)	0.7 (±0.13)	0.7 (±0.16)	0.18	0.74	0.67	0.63
Ether extract	0.05 (±0.01)	0.05 (±0.01)	0.05 (±0.01)	0.05 (±0.01)	0.05	0.75	0.48	0.53
Neutral detergent fiber	1.8 (±0.36)	1.8 (±0.53)	1.8 (±0.35)	1.8 (±0.44)	0.03	0.86	0.66	0.64
Acid detergent fiber	0.9 (±0.19)	1.0 (±0.27)	1.0 (±0.18)	1.0 (±0.23)	0.04	0.85	0.66	0.63
Digestibility, %								
Dry matter	74.5 (±1.59)	74.9 (±1.91)	75.3 (±1.82)	73.7 (±2.26)	0.64	0.50	0.14	0.46
Organic matter	77.8 (±1.53)	78.3 (±1.90)	78.6 (±1.83)	77.0 (±2.27)	0.62	0.46	0.12	0.48
Crude protein	80.0 (±2.64)	81.9 (±2.18)	82.4 (±2.71)	80.3 (±0.69)	0.41	0.70	0.03	0.75
Ether extract	86.8 (±0.88)	86.9 (±0.62)	86.0 (±1.81)	85.2 (±1.63)	0.31	0.06	0.50	0.69
Neutral detergent fiber	57.7 (±1.21)	56.9 (±4.48)	57.6 (±3.56)	54.2 (±4.13)	1.69	0.21	0.47	0.46
Acid detergent fiber	49.2 (±1.92)	48.2 (±4.96)	50.8 (±5.95)	44.8 (±5.26)	2.25	0.31	0.28	0.24

**Table 4 animals-12-02473-t004:** Effect of HY supplementation on rumen fermentation in growing beef cattle.

Item	HY, g/kg DM					Contrast		
	0	1	2	3	SEM	L	Q	C
pH	6.8 (±0.65)	6.9 (±0.36)	6.8 (±0.11)	6.9 (±0.44)	0.65	0.22	0.36	0.97
NH_3_-N, mg/dL	15.5 (±1.32)	18.0 (±2.31)	17.7 (±2.67)	18.3 (±1.58)	0.53	0.02	0.17	0.23
Total VFA, mmol/d	100.8 (±5.98)	104.5 (±4.13)	110.8 (±3.01)	106.0 (±2.12)	0.85	0.01	0.03	0.10
VFA, mol/100 mol								
Acetate (C2)	67.5 (±2.27)	67.3 (±3.92)	63.3 (±1.24)	66.0 (±2.09)	0.46	0.06	0.14	0.03
Propionate (C3)	22.6 (±0.71)	22.6 (±1.07)	24.4 (±0.90)	23.4 (±0.95)	0.17	0.03	0.23	0.01
Butyrate (C4)	9.9 (±2.41)	10.1 (±3.05)	12.3 (±0.98)	9.5 (±1.93)	0.28	0.80	0.03	0.02
C2:C3	3.0 (±0.17)	3.0 (±0.32)	2.6 (±0.14)	2.9 (±0.18)	0.04	0.09	0.13	0.01

NH_3_-N: amonia nitrogen; VFA: volatile fatty acids.

**Table 5 animals-12-02473-t005:** Effect of HY supplementation on rumen microorganisms in growing beef cattle.

Item	HY, g/kg DM				SEM	Contrast		
	0	1	2	3		L	Q	C
Microbial population, cell/mL								
Bacteria, ×10^10^	2.3 (±0.97)	2.3 (±1.06)	2.4 (±0.65)	2.4 (±1.02)	0.01	<0.01	0.78	0.05
Protozoa, ×10^5^	4.1 (±0.82)	4.3 (±1.15)	4.6 (±0.89)	4.3 (±1.04)	0.43	0.66	0.58	0.72
Fungi, ×10^4^	0.4 (±0.22)	0.3 (±0.27)	0.4 (±0.22)	0.7 (±0.45)	0.09	0.11	0.15	1.00

**Table 6 animals-12-02473-t006:** Effect of HY supplementation on hematology in growing beef cattle.

Item	HY (g/kg DM)					Contrast		
	0	1	2	3	SEM	L	Q	C
BUN, mg/dL	17.4 (±3.13)	18.4 (±3.29)	18.0 (±3.54)	18.4 (±3.36)	1.87	0.71	0.85	0.75
Red blood cell, 10^12^/L	5.6 (±0.57)	6.0 (±1.05)	6.2 (±1.48)	5.5 (±1.01)	1.03	0.97	0.28	0.73
Hemoglobin, g/dL	7.9 (±1.00)	8.4 (±1.34)	8.9 (±2.49)	7.9 (±1.57)	1.29	0.84	0.35	0.71
Hematocrit, %	23.8 (±3.03)	25.4 (±4.04)	26.8 (±7.43)	24.0 (±4.64)	2.28	0.84	0.34	0.69
White blood cells, 10^9^/L	10.7 (±1.25)	13.2 (±2.29)	11.2 (±2.73)	12.1 (±0.80)	4.61	0.62	0.41	0.11
Neutrophils, %	33.0 (±3.87)	35.4 (±9.40)	40.6 (±5.13)	46.4 (±8.14)	2.76	0.01	0.62	0.88
Lymphocytes, %	47.6 (±13.96)	56.4 (±10.45)	56.0 (±13.78)	44.0 (±9.87)	3.66	0.18	0.10	0.93
Monocytes, %	0.08 (±1.10)	2.20 (±1.30)	1.20 (±1.10)	3.00 (±1.41)	0.83	0.04	0.73	0.07
Eosinophils, %	7.2 (±3.42)	6.0 (±2.16)	6.8 (±3.16)	7.6 (±4.16)	2.10	0.82	0.62	0.82

## Data Availability

Not applicable.

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
