# Peer review of "Effect of Dietary Supplementation of Hydrolyzed Yeast on Growth Performance, Digestibility, Rumen Fermentation, and Hematology in Growing Beef Cattle"

_animals, 2022, doi:10.3390/ani12182473_

Round 1

Reviewer 1 Report

 If digestibility of crude protein and ether extract were increased with the HY, it should have been reflected on the performance of beef.

Author Response

Thank you very much for the editor and reviewer’s comments. I added the data for growth performance and hematology as your suggestion and also changed the title of this manuscript.

Responses to Reviewer:
Reviewer 1

Introduction

L72: There was no punctuation.

-L84: Already changed, please see in text.

Materials and methods

Results

  1. 1. L134: “CH4: methane.”,but there was no this item in the table.

-Already deleted, please see in text.

  1. 2. If digestibility of crude protein and ether extract were increased with the HY, it should have been reflected on the performance of beef.

-Table 3: I am reanalyzing statistical the digestibility of nutrients. Then ether extract showed no significant difference when added HY.

-Already added the data on the growth performance of beef cattle and changed in “Materials and methods, Section 3.1 and 4.1 Performance and Table 2”, please see in text.

Discussion

  1. 1. L186(This could be because the HY consists 186 of peptides, amino acids, and nucleotides to degrade for NH3-N production by rumen 187 microorganisms) was inconsistent with L184( increased NH3-N concentration in beef cattle fed the HY…).

-L218-224: Already changed to “The increased NH3-N concentration in beef cattle fed HY was consistent with the results reported by Öztürk et al. [44], the addition of HY improved the NH3-N concentration in the rumen simulation technique (Rusitec). Oeztuerk et al. (45) found that a HY stimulated the proteolytic activity of rumen bacteria, which in turn led to an increase in the concentration of NH3-N in Rusitec. In the current study, the greater ruminal NH3-N concentration can be attributed to the microbial breakdown of yeasts due to their high protein content.”, please see in text.

  1. 2. L213 : the focus of this discussion about microbial population was cellulolytic bacterial species, but the result was increased NH3-N concentration in beef cattle fed the HY, it should have been reflected on microbial population related to NH3-N.

-L245-248: Already changed to “Most rumen bacteria are capable of utilizing ammonia as their major nitrogen source [53]. High NH3-N concentration when adding HY can enhance the bacterial population in cattle due to effective bacterial biosynthesis in the rumen.”, please see in text.

  1. 3. L210 (R:C ratio) was inconsistent with L215(forage to concentrate 215 ratio…).

-Already deleted, please see in text.

Conclusion

The Conclusion was the HY supplementation at 2 g/kg DM improved bacterial population…, but there was no detailed exposition in the part of “Results”.

-L 270-274: Already changed in conclusion to “Dietary supplementation of HY enhanced digestibility of CP and hematological indices, especially neutrophils and monocytes in growing beef cattle. In addition, supplementation of HY enhanced total VFA and propionate production. However, supplementation of HY had no effect on the growth performance. The HY was fed to growing beef cattle at a dose of 2 g/kg DM as a rumen modifier and to improve health status.”, please see in text.

-L154-163: Already changed in the results to “The rumen pH was not significantly different among treatments (p > 0.05), while rumen NH3-N linearly increased (p = 0.02) when increasing level of HY (Table 4).  The concentration of total VFA was increased quadratically (p = 0.03) when cattle were fed with HY supplementation. The proportions of propionate (C3) were increased cubically (p = 0.01), while acetate (C2) and C2:C3 were decreased cubically (p < 0.05) by HY supplementation. The butyrate proportion was cubically decreased (p = 0.02) by HY addition. Furthermore, total VFA, propionate and butyrate were highest, while acetate was lowest in cattle fed with the addition of HY at 2 g/kg DM. The supplementation of HY did not influence the protozoal and fungal population (p > 0.05) (Table 5). However, the bacterial population cubically increased (p < 0.01) with supplementation of HY.”, please see in text.

Reviewer 2 Report

The study entitled: Hydrolyzed Yeast Supplement Improves Nutrient Digestibility, Rumen Fermentation and Bacterial Population in Brahman × Thai Native Crossbred Beef Cattle….However this study is  simple and has not advanced techniques particularly those related with rumen microbial ecosystem and methane emission. It is generally provide information on a new relatively feed additive that can be used as a natural growth promoter for food producing animals. The major concern in this study is related to the animals performance, as no data on growth performance or feed efficiency were provided. What is the return of the manipulation of the rumen ecosystem if this was not reflected on growth, meat production. The emphasis on the mode of action of this product on rumen ecosystem and growth performance of animals should be provided along with the following comments:

Simple summary: this part should be rewritten. It contains incomplete sentences for example : The aim of this study is to determine the different levels of hydrolyzed  yeast supplementation in crossbred beef cattle….Is not informative and does not show the aim of the study.

Introduction:

More details on the mode of action of this supplement is required. For example, how the cell wall of yeast and/or other ingredients, active components, can modulate rumen ecosystem and nutrient digestibility.

Abstract and results: the digestibility of crude protein and ether extract were increased (p < 0.05) by HY supplementation….Please, revise you have to show the type of increase, linear or cubic according to the results shown in Table 3. These relationships have different biological meanings and should be clearly stated and explained in discussion section.  

Line 63: diet addition with HY replace addition with supplemented

Materials and methods

Line 83: The cattle were divided into five blocks according to sex….this statement is not clear… how animals were divided into blocks according to sex?????? if you mean that females and males were used in this study. The effect of the sex should be inserted as fixed effect in the model. To me, I cannot understand how animals divided into blocks according to sex. The block usually refers to the pen and as stated animals were individually housed.

Line 84: ad libitum, What do you mean by ad libitum?. Usually, this means that the feed is offered in front of animals all time for free consumption. How this was done while determining feed intake? Please, explain.

Line 91:. Using anal stimulation, fresh fecal samples (about 500 g) were taken from each 91

animal every morning for five consecutive days. …. I am not sure if collecting feces for five consecutive day, not clear at which time of the experimental period, can be representative samples to the digestibility trail.

Line 98: At 4 hours after feeding on the final day of the experiment, 200 ml of rumen fluid 98

was collected using a stomach tube attached to a vacuum pump…..Is this method still acceptable? Fastulated animals are more suitable???

Statistical analysis section:

Please, show in statistical analysis on which base you select the significance of the contrast regression analysis when more than one regression give p value less than 0.05.

Line 127:  The pro-portion of total VFA, propionate (C3) and butyrate (C4) were increased (p < 0.05), while

Line 129: acetate (C2) and C2:C3 were decreased cubically (p < 0.05) by HY supplementation….revise the statement the pattern of increase in butyrate is not similar to total VFA ans propionate.

Section 2.3.

Please, provide more detailed procedures.

Please check, it may help:

Mousa G.A., Allak M.A., Shehata M.G., Hashem N.M., Hassan O.G.A. 2022. Dietary Supplementation with a Combination of Fibrolytic Enzymes and Probiotics Improves Digestibility, Growth Performance, Blood Metabolites, and Economics of Fattening Lambs. Animals, 12, 476. https://doi.org/10.3390/ani12040476.

Soltan, Y. A., Morsy, A. S., Hashem, N. M., Sallam, S. M. 2021. Boswellia sacra resin as a phytogenic feed supplement to enhance ruminal fermentation, milk yield, and metabolic energy status of early lactating goats. Animal Feed Science and Technology, 114963.

Table 4: delete CH4 at the footnote, there is no CH4 determination in this study.

Discussion:

The discussion mainly depends on the comparison between previous studies and the present study. Again, more focus on the explanation of the study results are required, considering the results of the regression analysis. Some variables were cubically or quadratically  affected. This needs deeper discussion.

Line 183: the increase in ammonia level in the rumen is not always ggod indicator, this increase may be due to diet protein hydrolysis, which is not required. Also, high ammonia if is not associated with sufficient energy source, carbohydrate, may drive to negative impacts on animal health. Please, consider this in the discussion.

Author Response

Thank you very much for the editor and reviewer’s comments. I added the data for growth performance and hematology as your suggestion and also changed the title of this manuscript.

Reviewer 2

The study entitled: Hydrolyzed Yeast Supplement Improves Nutrient Digestibility, Rumen Fermentation and Bacterial Population in Brahman × Thai Native Crossbred Beef Cattle….However this study is  simple and has not advanced techniques particularly those related with rumen microbial ecosystem and methane emission. It is generally provide information on a new relatively feed additive that can be used as a natural growth promoter for food producing animals. The major concern in this study is related to the animals performance, as no data on growth performance or feed efficiency were provided.

-Already added the data on the growth performance of beef cattle and changed in “Materials and methods, Section 3.1 and 4.1 Performance and Table 2”, please see in text.

What is the return of the manipulation of the rumen ecosystem if this was not reflected on growth, meat production. The emphasis on the mode of action of this product on rumen ecosystem and growth performance of animals should be provided along with the following comments:

Simple summary: this part should be rewritten. It contains incomplete sentences for example : The aim of this study is to determine the different levels of hydrolyzed  yeast supplementation in crossbred beef cattle….Is not informative and does not show the aim of the study.

-L27-29: Already changed to “The aim of this study was to determine the different levels of hydrolyzed yeast supplementation on feed utilization, rumen fementation, hematology and growth performance in growing beef cattle.”, please see in text.

Introduction:

More details on the mode of action of this supplement is required. For example, how the cell wall of yeast and/or other ingredients, active components, can modulate rumen ecosystem and nutrient digestibility.

-L70-80: Already changed to “Yeast cell walls are composed of β-glucans, mannan-oligosaccharides (MOS), and yeast extracts such as nucleotide and amino acid profile [12,13]. These polysaccharides prevent phatogenic bacteria in the digestive tract and interact directly with immune cells [1]. HY addition improves anti-bacterial immunity mediated by macrophages and neutrophils [14]. HY provides various growth factors, vitamins, and other nutrients to promote the development of rumen microorganisms, especially bacteria [15]. It has been found that yeast stimulates lactate-utilizing bacteria to use lactic acid (16) and enhances the cellolytic bacterial population, thereby increasing the digestion of fiber [17,18]. Additionally, it also stimulates fermentation and improves rumen VFA production, especially propionate [19], which may enhance growth performance in ruminant.”, please see in text.

Abstract and results: the digestibility of crude protein and ether extract were increased (p < 0.05) by HY supplementation….Please, revise you have to show the type of increase, linear or cubic according to the results shown in Table 3. These relationships have different biological meanings and should be clearly stated and explained in discussion section.

-Table 3: I am reanalyzing statistical the digestibility of nutrients. Then ether extract showed no significant difference when added HY.

-L40-41 (abstract): Already changed to “………while the digestibility of crude protein (CP) increased quadratically (p = 0.03) in the cattle receiving HY.”, please see in text.

-L149-50 (results): The digestibility of CP increased quadratically (p = 0.03) with increasing HY supplementation……”, please see in text.

Line 63: Already changed to “diet addition with HY replace addition with supplemented

-Already deleted this sentences, please see in text.

Materials and methods

Line 83: The cattle were divided into five blocks according to sex….this statement is not clear… how animals were divided into blocks according to sex?????? if you mean that females and males were used in this study. The effect of the sex should be inserted as fixed effect in the model. To me, I cannot understand how animals divided into blocks according to sex. The block usually refers to the pen and as stated animals were individually housed.

-1/2:

Thank reviewer very much for suggesting this error. The authors agree that the statement is not clear and needs more information about sex. We therefore rewrite, as follows:

L34-37 (Abstract): Already changed to “Twenty crossbred beef cattle with an initial body weight (BW) of 142±12 kg were randomly assigned to one of four treatments for 90 days in a randomized complete block design (RCBD) having five blocks based on a homogenous subpopulation of sex and BW.”, please see in text.

L92-98: Already changed to “In this study, a feeding trial (90-d) to compare dietary hydrolyzed yeast (HilysesTM, ICC, Sao Paulo, Brazil) supplementation at 0, 1, 2 and 3 g/kg DM were conducted using a randomized complete block design (RCBD). Twenty crossbred (Brahman x Thai native) yearling beef cattle, consisting of sixteen males and four females with 142±12 kg of BW (mean ± SD), were grouped into five blocks according to sex and their homogenous BW, respectively. The animals within each block were assigned randomly to one of four dietary treatments.”, please see in text.

-2/2:

The nutrient utilization between males and females of beef cattle during growing stage is quite similar. In this study, the part of females was only one replication/treatment. Therefore, transformation of model to support the effects of sex should be not applicable.

We understand that the block usually refers to the pen, animal factor (i.e. breed, sex, age, and BW) as well as location. In tradition, seldom experiments use sex or sex-BW mixture as a block. Based on every standard RCBD, the dissimilarity of experimental units among blocks is not able to specify although it plays important roles to reduce the potential bias(es) from treatment means. Concept of RCBD might be that the experimental units must be homogenous within block or replicate. There was a published reference (Leheska et al., 2009; Kongphitee et al., 2018), who used sex-BW mixture as a block. Therefore, we believe that our experimental design was a type of RCBD.

Ref:

  1. Leheska, J.M.; Montgomery, J.L.; Krehbiel, C.R.; Yates, D.A.; Hutcheson, J.P.; Nichols, W.T.; Streeter, M.; Blanton Jr., J.R.; Miller, M.F. 2009. Dietary zilpaterol hydrochloride. II. Carcass composition and meat palatability of beef cattle. Journal of Animal Science. 87, 1384-1393. doi:10.2527/jas.2008-1168.
  2. Kongphitee, K.; Sommart, K.; Phonbumrung, T.; Gunha, T.; Suzuki, T. 2018. Feed intake, digestibility and energy partitioning in beef cattle fed diets with cassava pulp instead of rice straw. Asian Australisian Journal of Animal Science, 31, 1431–1441. https:// doi.org/10.5713/ajas.17.0759.

Line 84: ad libitum, What do you mean by ad libitum?. Usually, this means that the feed is offered in front of animals all time for free consumption. How this was done while determining feed intake? Please, explain.

- L98-99: Already changed to “The cattle were provided TMR (Table 1) ad libitum, targeting a refusal of 10% on an as-fed basis and fed twice a day at 08:00 h and 17:00 h.”, please see in text.

Line 91:. Using anal stimulation, fresh fecal samples (about 500 g) were taken from each animal every morning for five consecutive days. …. I am not sure if collecting feces for five consecutive day, not clear at which time of the experimental period, can be representative samples to the digestibility trail.

-L107-110: Already changed to “At the end of the experiment, a digestibility test was conducted in which cattle were housed in individual pens for five consecutive days (86 to 90-d of the trial). Rectal sampling was used to collect fresh fecal samples (about 500 g). The composite samples of each cattle's daily fresh feces were combined and then refrigerated.”, please see in text.

Line 98: At 4 hours after feeding on the final day of the experiment, 200 ml of rumen fluid was collected using a stomach tube attached to a vacuum pump…..Is this method still acceptable? Fastulated animals are more suitable???

This method suitable for rumen sampling. Many experiment use evaluate for rumen fermentation of the animals such as:

  1. Gunun, N.; Ouppamong, T.; Khejornsart, P.; Cherdthong, A.; Wanapat, M.; Polyorach, S.; Kaewpila, C.; Kang, S.; Gunun, P. 2022. Effects of Rubber Seed Kernel Fermented with Yeast on Feed Utilization, Rumen Fermentation and Microbial Protein Synthesis in Dairy Heifers. Fermentation 8, 288.
  2. de Assis Lage, C. F., S. E. Räisänen, A. Melgar, K. Nedelkov, X. Chen, J. Oh, M. E. Fetter, N. Indugu, J. S. Bender, B. Vecchiarelli, M. L. Hennessy, D. Pitta, and A. N. Hristov. 2020. Comparison of two sampling techniques for evaluating ruminal fermentation and microbiota in the planktonic phase of rumen digesta in dairy cows. Front. Microbiol. 11, 618032.
  3. Cherdthong, A.; Khonkhaeng, B.; Foiklang, S.;Wanapat, M.; Gunun, N.; Gunun, P.; Chanjula, P.; Polyorach, S. 2019. Effects of supplementation of Piper sarmentosum leaf powder on feed e_ciency, rumen ecology and rumen protozoal concentration in Thai native beef cattle. Animals 9, 130.
  4. Song, J.; Choi, H.; Jeong, J. Y.; Lee, S. ; Lee, H. J.; Baek, Y.; Ji, S.Y.; Kim, M. 2018. Effects of sampling techniques and sites on rumen microbiome and fermentation parameters in hanwoo steers. J. Microbiol. Biotechnol. 28, 1700–1705.

Statistical analysis section:

Please, show in statistical analysis on which base you select the significance of the contrast regression analysis when more than one regression give p value less than 0.05.

L140-141: We thank to reviewer for this opinion. We insert the statement. Already changed to “When the contrasts were significant, the effect was based on the higher level, i.e. cubic, quadratic, and linear, respectively.”, please see in text.

Line 127:  The pro-portion of total VFA, propionate (C3) and butyrate (C4) were increased (< 0.05), while

Line 129: acetate (C2) and C2:C3 were decreased cubically (p < 0.05) by HY supplementation….revise the statement the pattern of increase in butyrate is not similar to total VFA ans propionate.

-L155-160: Already changed to “The concentration of total VFA was increased quadratically (p = 0.03) when cattle were fed with HY supplementation. The proportions of propionate (C3) were increased cubically (p = 0.01), while acetate (C2) and C2:C3 were decreased cubically (p < 0.05) by HY supplementation. Furthermore, total VFA, propionate were highest, while acetate was lowest in cattle fed with the addition of HY at 2 g/kg DM. The butyrate proportion was cubically decreased (p = 0.02) by HY addition.”, please see in text.

Section 2.3.

Please, provide more detailed procedures.

Please check, it may help:

Mousa G.A., Allak M.A., Shehata M.G., Hashem N.M., Hassan O.G.A. 2022. Dietary Supplementation with a Combination of Fibrolytic Enzymes and Probiotics Improves Digestibility, Growth Performance, Blood Metabolites, and Economics of Fattening Lambs. Animals, 12, 476. https://doi.org/10.3390/ani12040476.

Soltan, Y. A., Morsy, A. S., Hashem, N. M., Sallam, S. M. 2021. Boswellia sacra resin as a phytogenic feed supplement to enhance ruminal fermentation, milk yield, and metabolic energy status of early lactating goats. Animal Feed Science and Technology, 114963.

-L104-132: Already changed in section 2.3. Data collection and sampling procedures, please see in text.

Table 4: delete CH4 at the footnote, there is no CH4 determination in this study.

-Already deleted, please see in text.

Discussion:

The discussion mainly depends on the comparison between previous studies and the present study. Again, more focus on the explanation of the study results are required, considering the results of the regression analysis. Some variables were cubically or quadratically  affected. This needs deeper discussion.

-Thank you for your suggestion. We try to explanation of the study results, please see in “Discussion”.

Line 183: the increase in ammonia level in the rumen is not always good indicator, this increase may be due to diet protein hydrolysis, which is not required. Also, high ammonia if is not associated with sufficient energy source, carbohydrate, may drive to negative impacts on animal health. Please, consider this in the discussion.

-L224-226: Already changed to “The concentration of rumen NH3-N at 4 hours after feeding ranges from 15.5 to 18.3 mg/dL, which is close to the optimum level (13 to 24 mg/dL) for microbial growth in the rumen as recommended by

  1. Wanapat, M.; Pimpa, O. Effect of ruminal NH3-N levels on ruminal fermentation, purine derivatives, digestibility and rice straw intake in swamp buffaloes. Asian-Aus. J. Anim. Sci. 1999, 12, 904-907.
  2. Supapong, C.; Cherdthong, A.; Wanapat, M.; Chanjula, P.; Uriyapongson, S. Effects of sulfur levels in fermented total mixed ration containing fresh cassava root on feed utilization, rumen characteristics, microbial protein synthesis, and blood metabolites in Thai native beef cattle. Animals 2019, 9, 261.
  3. Gunun P.; Wanapat M.; Gunun, N.; Cherdthong, A.; Sirilaophaisan, S.; Kaewwongsa W. Effects of condensed tannins in mao (Antidesma thwaitesianum Muell. Arg.) seed meal on rumen fermentation characteristics and nitrogen utilization in goats. Asian-Australas. J. Anim. Sci. 2016, 29, 1111-1119.
  4. Gunun, N.; Ouppamong, T.; Khejornsart, P.; Cherdthong, A.; Wanapat, M.; Polyorach, S.; Kaewpila, C.; Kang, S.; Gunun, P. Effects of rubber seed kernel fermented with yeast on feed utilization, rumen fermentation and microbial protein synthesis in dairy heifers. Fermentation 2022, 8, 288.

In addition, the inclusion of HY had no effect on BUN concentration and it ranged from 17.4-18.4 mg/dl was within the usual range of 10 to 20 mg/dl [54,55].” This study showed that NH3-N and BUN did not affect on health status of cattle.

Reviewer 3 Report

Dear Authors,

Thank you for submitting this manuscript that investigates the use of hydrolyzed yeast supplements for improving nutrient digestibility in cattle. There is a clear incentive to studies of this sort, and there could be considerable interest from those in the agriculture sector regarding the results.

At current however, there seem to be some large revisions required in the manuscript to ensure the work is scientifically robust. I have attached the PDF version of the manuscript with specific comments. Additionally, please consider the following points: 

1. Introduction. This is very brief and does not explain the application and potential benefits of HY well. There is some misleading information regarding antibiotic use in ruminants.

2. Methods. Please ensure these are explained fully. Aspects of animal husbandry (where and how are they kept) and food presentation (pellet, mash etc) all could impact the results and as such, need to be explained clearly for repeatability.

3. Test results. Please ensure the test statistics are provided and the actual p values are given. This is essential from a scientific standpoint. Please also check the points relating to test choice and normal distribution.

4. Wording. There are some grammatical errors in the work that make some sentences misleading. Please review the work thoroughly for spelling and grammar errors.

With these revisions, the work should be in a stronger position for consideration.

Author Response

Thank you very much for the editor and reviewer’s comments. I added the data for growth performance and hematology as your suggestion and also changed the title of this manuscript.

Reviewer 3

Dear Authors,

Thank you for submitting this manuscript that investigates the use of hydrolyzed yeast supplements for improving nutrient digestibility in cattle. There is a clear incentive to studies of this sort, and there could be considerable interest from those in the agriculture sector regarding the results.

At current however, there seem to be some large revisions required in the manuscript to ensure the work is scientifically robust. I have attached the PDF version of the manuscript with specific comments. Additionally, please consider the following points: 

  1. Introduction. This is very brief and does not explain the application and potential benefits of HY well. There is some misleading information regarding antibiotic use in ruminants.

-L58-64: Already changed to “Antibiotics have been administered for the prevention and treatment of disease in animals, plants, and humans [4]. However, a large portion of the antibiotics generated each year around the world are utilized for non-therapeutic uses [5]. In addition, antibiotics have been utilized as growth promoters and feed enhancers, and not for the treatment of disease [6-8]. However, given the prevalence of antibiotic resistance and concern about its future global impact, research into ways to support antibiotic restriction is critical [9].”, please see in text.

  1. Methods. Please ensure these are explained fully. Aspects of animal husbandry (where and how are they kept) and food presentation (pellet, mash etc) all could impact the results and as such, need to be explained clearly for repeatability.

L85-132: Already modified in Materials and Methods, please see in text.

  1. Test results. Please ensure the test statistics are provided and the actual p values are given. This is essential from a scientific standpoint. Please also check the points relating to test choice and normal distribution.

-1/2: Thank reviewer. The authors have checked the actual p values entire text according to your recommendation. The manuscript is revised accordingly.

L154-163: Already changed to“The rumen pH was not significantly different among treatments (p > 0.05), while rumen NH3-N linearly increased (p = 0.02) when increasing level of HY (Table 4). The concentration of total VFA was increased quadratically (p = 0.03) when cattle were fed with HY supplementation. The proportions of propionate (C3) were increased cubically (p = 0.01), while acetate (C2) and C2:C3 were decreased cubically (p < 0.05) by HY supplementation. Furthermore, total VFA, propionate were highest, while acetate was lowest in cattle fed with the addition of HY at 2 g/kg DM. The butyrate proportion was cubically decreased (p = 0.02) by HY addition. The supplementation of HY did not influence the protozoal and fungal population (p > 0.05) (Table 5). However, the bacterial population cubically increased (p < 0.01) with supplementation of HY.”, please see in text.

-2/2:

In this study, the block was statistically significant (p < 0.05). Therefore, the candidate of ANCOVA procedure is not applicable. Before the ANOVA procedure (drawn as GLM), the data were tested normal distribution by using PRINT procedure. In addition, no outliers were detected with BOX PLOT analysis.

  1. Wording. There are some grammatical errors in the work that make some sentences misleading. Please review the work thoroughly for spelling and grammar errors.

-I have already checked and improved grammar, please see in text. 

With these revisions, the work should be in a stronger position for consideration.

Round 2

Reviewer 2 Report

The article was corrected according to the reviewers comments.

Author Response

The article was corrected according to the reviewers comments.

-Thank you for your recommendation.

Reviewer 3 Report

Dear Authors,

Thank you for providing a revised version of your manuscript. While some of the key points from the original review have been addressed, many have not (e.g. use of scientific names). Additionally, the work is still confusingly written with many sentences that do not make sense. Please ensure a full proof read by a native english speaker is conducted, and all points on the attached PDF file have been addressed.

Author Response

Thank you for providing a revised version of your manuscript. While some of the key points from the original review have been addressed, many have not (e.g. use of scientific names). Additionally, the work is still confusingly written with many sentences that do not make sense. Please ensure a full proof read by a native english speaker is conducted, and all points on the attached PDF file have been addressed.

- Thank you for your suggestions and already improved English sentences, which can be found in the text. 

Include the scientific name on the first mention.

-L33: Already changed to “….in growing crossbred Bos indicus cattle.”, which can be found in the text. 

Report the actual p-value here.

- Already added in the “Abstract” and “Results”, which can be found in the text. 

Again, make sure that the actual p value is provided in the work.

-L43: Already changed to “The proportion of acetate decreased cubically (p = 0.03)”, which can be found in the text.

Please review the grammer here.

-Already deleted.

Some of the key words are already included in the title. Remove any key words that are in the title and use new terms to increase paper discoverability

-L54: Already changed to “Keywords: hydrolyzed yeast; average daily gain; propionate; bacterial population, hematological parameters”, which can be found in the text.

Most important ......?

-L58-60: Already changed to “Farmers, feed manufacturers, and animal nutritionists are becoming highly interested in feed additives to improve the feed utilization, rumen microbial fermentation, health, and performance of their animals in tropical areas [1-3].”, which can be found in the text.

Ruminants?

-L60-61: Already changed to "Antibiotics have been administered for the prevention and treatment of disease in animals, plants, and humans [4]", which can be found in the text.

There are also some major concerns for ruminant health when using antibiotics, given their impact on commensal bacteria. There needs to be a much clearer explanation on the appropriate use of antibiotics and their purported benefits.

-L60-66: Antibiotics have been used for the prevention and treatment of disease in animals, plants, and humans [4]. However, a large portion of the antibiotics generated each year around the world are utilized for non-therapeutic purposes [5]. In addition, antibiotics have been utilized as growth promoters and feed enhancers, and not for the treatment of disease [6-8]. However, given the prevalence of antibiotic resistance and concern about its future global impact, research into ways to support antibiotic restriction is critical [9].”, which can be found in the text.

Sentence doesn't make sense. Please rephrase.

-L72-74: Already changed to “HY is the whole yeast residue remaining after the lysis process and contains β-glucans, mannan-oligosaccharides (MOS), nucleotides, peptides, and amino acids [10,12,13].”, which can be found in the text.

The introduction is currently brief and does not explain the background to the study well. More information is needed.

-Already changed and explain the background of the study, please see in the "Introduction".

Use the scientific name when first mentioning a species.

Please use a full stop here.

-L86: Already changed, please see in text.

Was the sex ratio equal?

-Male to female ratio at 4:1.

Do you mean as in one animal per pen?

Please provide more information on the conditions for cattle. Were they given access to grazing? It is essential this is explained because it could affect results.

-L104-105: Already changed to “The cattle were kept in separate pens all the time with access to water and mineral blocks.”, which can be found in the text.

More information is needed on diet preparation. is this in pellet form? How was the supplement added? Are all pellets of the same size, and if so, what size?

-L100-103: Already changed to “The total mixed ration (TMR) was prepared with 40% rice straw and 60% concentrate mixtures. Rice straw was chopped (4 cm. length) by machine (JrFarm, Nakhon Ratchasima, Thailand). Chopped rice straw was mixed with concentrate ingredients to prepare TMR.”, which can be found in the text.

At what temperature? For how long? If it is warm, the bacteria may continue to generate VFAs.

- L114-115: Already changed to “The composite samples of each cattle's daily fresh feces were combined and then refrigerated at 4 °C.”, which can be found in the text.

This method needs to be explained more clearly.

-L124-126: Already changed to “In addition, 1 ml of rumen fluid was diluted with 9 ml of formalin-saline solution. The numbers of bacteria, protozoa, and fungi were counted with a haemacytometer (Boeco, Hamburg, Germany) by the Galyean method [23]. ”, which can be found in the text.

This is a parametric test and assumes normal distribution of data. Did you test your data for normality and if so what was the finding? Please report it here. If the data are not parametric an alternative test (such as Kruskal Wallis) would be more appropriate.

-L139-140: Already changed to “All data were tested for normal distribution using the UNIVARIATE procedure in SAS software and subjected to analysis of variance using the GLM procedure [27].”, which can be found in the text.

Please make sure you include the test statistic and the actual p value in all cases. This is important for repeatabiliy of the study.

-We revised the result section with the actual P value accordingly.

Please include a measure of variance, such as standard deviation, here.

- Already included standard deviation in all table.

As previous - include test statistics and the actual p values in all cases.

- Already included, which can be found in the text.

Measure of variance - standard deviation?

- Already include standard deivation in all table.

How these were identified is poorly explain in the methods.

-L124-126: Already cahged to “In addition, 1 ml of rumen fluid was diluted with 9 ml of formalin-saline solution. The numbers of bacteria, protozoa, and fungi were counted with a haemacytometer (Boeco, Hamburg, Germany) by the Galyean method [23]. ”, which can be found in the text.”, which can be found in the text.

Wording doesn't make sense here. Could you rephrase?

-L199-200: Already changed to “Hence, the addition of HY to the cattle enhanced the digestibility of CP.”, which can be found in the text.

Cattle feeds?

-L200: Already changed to “Hence, the addition of HY to the cattle …………………..”, which can be found in the text.

Very brief conclusion. Could this be developed further?

-L277-283: Already changed to “Dietary supplementation of HY enhanced the digestibility of CP and hematological indices, especially neutrophils and monocytes in growing beef cattle. In addition, supplementation of HY enhanced total VFA and propionate production. However, supplementation of HY did not affect the growth performance. The HY was fed to growing beef cattle at a dose of 2 g/kg DM as a rumen modifier and to improve health status. Therefore, further studies are required to determine the effect of HY on carcass traits and meat quality in beef cattle.”, which can be found in the text.

References are well formatted to the MDPI style. There are just a few small inconsistencies. Check through page numbers for journals as there are many cases where only one page number is reported.

-Already used references in the format of the MDPI style and checked all page numbers. In addition, the journal under the MDPI had only one page.

Pages?

-L335: Page 119.

No need to include year here.

-Already deleted this reference.

Pages

-L376: Page 53.
